# Biphasic response as a mechanism against mutant takeover in tissue homeostasis circuits

Omer Karin 🆔 & Uri Alon*🆔

## Abstract

Tissues use feedback circuits in which cells send signals to each other to control their growth and survival. We show that such feedback circuits are inherently unstable to mutants that misread the signal level: Mutants have a growth advantage to take over the tissue, and cannot be eliminated by known cell-intrinsic mechanisms. To resolve this, we propose that tissues have biphasic responses in and the signal is toxic at both high and low levels, such as glucotoxicity of beta cells, excitotoxicity in neurons, and toxicity of growth factors to T cells. This gives most of these mutants a frequency-dependent selective disadvantage, which leads to their elimination. However, the biphasic mechanisms create a new unstable fixed point in the feedback circuit beyond which runaway processes can occur, leading to risk of diseases such as diabetes and neurodegenerative disease. Hence, glucotoxicity, which is a dangerous cause of diabetes, may have a protective anti-mutant effect. Biphasic responses in tissues may provide an evolutionary stable strategy that avoids invasion by commonly occurring mutants, but at the same time cause vulnerability to disease.

**Keywords** calcium homeostasis; design principles; evolutionary dynamics; mathematical models of disease; stem-cell homeostasis; tissue homeostasis

**Subject Categories** Quantitative Biology & Dynamical Systems; Signal Transduction

**Mol Syst Biol. (2017) 13: 933**

## Introduction

Maintaining proper tissue size is a fundamental problem for multicellular organisms. To do so, cells must precisely coordinate their proliferation and death rates, because an imbalance in these rates leads to either excessive growth or degeneration. Moreover, cells must coordinate their growth in the face of fluctuations, such as injury, or changes in the target size of the tissue, such as during development. This coordination requires feedback control.

Feedback circuits for controlling tissue size regulate cell growth by a signal that is affected by the size of the tissue. Thus, when the tissue is too small, the growth rate is positive, and when it is too large, the growth rate is negative. Only when the tissue reaches a desired size are the proliferation and death rates equal and the system reaches steady state. An example of such a feedback is the control of the concentration of T cells by IL-2 produced by the T cells (Hart *et al*, 2014). Another example occurs in endocrine tissues whose growth is regulated by the physiological variable that they control (Karin *et al*, 2016), such as in the control of beta-cell mass by blood glucose. These physiological feedback circuits can show dynamical compensation mechanisms that make them robust with respect to variation in parameters, such as insulin resistance in the case of glucose control (Karin *et al*, 2016). Feedback is also found in tissues that are renewed by proliferating stem cells (Lander *et al*, 2009), such as skeletal muscle and olfactory epithelium.

For such feedback circuits to function properly, each cell must respond precisely to the input signal. These responses depend on the activity and expression of receptors, signaling pathways, and regulatory proteins and are thus susceptible to mutations. When a mutant cell with a dysregulated proliferative or apoptotic response arises, it may invade the population and thus break the homeostatic control. Mutant takeover leads to aberrant tissue size and function. Thus, mechanisms must be in place to prevent such takeover.

One mechanism for protection from mutant invasion is cell intrinsic and concerns the paradoxical activation of apoptosis by c-myc (Lowe *et al*, 2004). C-myc is a transcription factor that drives proliferation in many cell types (Bouchard *et al*, 1998), yet it paradoxically induces apoptosis when overexpressed (Evan *et al*, 1992). This paradoxical induction of apoptosis plays an important role in tumor suppression because it eliminates transformed cells (Harrington *et al*, 1994; Lowe *et al*, 2004).

Here, we extend the idea of cell-intrinsic elimination of mutants to the level of circuits of communicating cells. We show that tissue feedback circuits are inherently sensitive to takeover by common types of mutants that misread the feedback signal, such as receptor loss-of-function or receptor locked-on mutations. The feedback loop gives these mutants a growth advantage relative to wild-type cells. We propose that mutant invasion can be prevented by a biphasic response mechanism, in which the signal is toxic to the cells at both low and high levels. Biphasic control gives the mutants a selective disadvantage compared to wild-type cells, and the mutants are hence eliminated.

Biphasic control of growth is prevalent in physiological systems. Examples include the control of beta-cell mass by glucose

Department of Molecular Cell Biology, Weizmann Institute of Science, Rehovot, Israel
*Corresponding author. Tel: +972 8 934 4448; E-mail: uri.alon@weizmann.ac.il

(Robertson *et al*, 2003), the control of mammary gland mass by estrogen (Lewis-Wambi & Jordan, 2009), the control of neuronal survival by glutamate (Hardingham & Bading, 2003), epidermal growth factor signaling (Högnason *et al*, 2001), and the control of T-cell concentration by IL2 and by antigen level (Critchfield *et al*, 1994; Hart *et al*, 2014). Biphasic control was also demonstrated for mechanical signaling—the control of epithelial cell proliferation by mechanical stretch through Piezo1 (Gudipaty *et al*, 2017). In all of these cases, signal is toxic at both high and low levels.

As mentioned above, we find that biphasic control can protect the tissue from invasion by commonly occurring mutants that mis-sense the feedback signal. However, we show that this protective mechanism comes at a cost. The biphasic response introduces an unstable fixed point. If the input signal fluctuates beyond this unstable fixed point, a runaway phenomenon occurs in which cells are eliminated and signal diverges, potentially leading to disease. We discuss this tradeoff between stability to mutant invasion and risk of disease in several systems, including beta-cell control of glucose, parathyroid control of calcium, stem-cell differentiation, and excito-toxicity in neurons.

# Results

### Tissue homeostasis circuits are inherently vulnerable to invasion by sensing mutants

Feedback circuits that control tissue size act to balance the proliferation and removal rates of the cells. The cells adjust their growth rate (proliferation minus death/removal) as function of an input $y$, which in turn is affected by the size of the tissue (Figs 1A and B, and EV1A and B).

There are two possible cases. In the first case, the signal $y$ increases with tissue size $Z$ (i.e., $Z$ activates $y$), and $y$ inhibits the growth rate of the cells (Figs 1A and EV1A). If there are too many cells, $y$ is large and growth rate is negative leading to reduction in tissue size. If there are too few cells, the opposite occurs and the tissue grows. This feedback loop guides the tissue to steady state at the point where growth rate is zero, at $y = y_{ST}$.

In the second case, the signal $y$ decreases with tissue size ($Z$ inhibits $y$) and $y$ increases the growth rate of the cells (Figs 1B and EV1B). The same considerations show that the tissue stably settles at $y = y_{ST}$. These feedback circuits thus provide a stable tissue size, because different initial cell populations and different initial concentrations of $y$ all converge on the same final population $Z_{ST}$ (Fig 1C). At the same time, the circuits also provide a stable signal level $y = y_{ST}$.

We propose that such generic feedback circuits are susceptible to invasion by mutants that misread the signal. When such a mutant arises at steady state, it senses the actual signal level $y_{ST}$ as a larger or smaller value $y_{MUT}$. In the case of Fig 1A, if the mutation causes a misreading of the signal as too low, as for example in a receptor-inactivating mutation, the feedback loop provides the mutant with a growth advantage and the mutant will take over the population. As a result, the tissue will show aberrant growth and, when the mutant is at high enough frequency, will show a level of $y$ that is too high, $y > y_{ST}$.

In the case of Fig 1B, if the mutation causes a misreading of the signal as too high, as for example in a locked-on receptor mutation,

the feedback loop provides the mutant with a growth advantage (Fig EV2A). The mutant will take over the population (Fig 1D). As a result, the tissue will show aberrant growth and, when the mutant is at high enough frequency, will show a level of $y$ that is too low, $y < y_{ST}$ (because in this case the tissue acts to reduce $y$). In both cases, sensing mutants can take over the population and only reach equilibrium again at an aberrant tissue size, leading to a breakdown of homeostatic control. Importantly, the same conclusion holds whether the growth of the cells is modeled as logistic or exponential (see Appendix Section S1), and when $y$ acts in delay (see Appendix Section S2).

### Biphasic response can protect against mutant invasion but can cause vulnerability to disease

To overcome the problem of mutant invasion, the sensing mutants need to have a selective disadvantage. One way to do this is an alternative implementation of the feedback circuit, in which $y$ affects the growth rate of $Z$ in a *biphasic* manner (Figs 1E and F, and EV1C and D). The word biphasic means that the growth rate curve has an inverse-U shape, with a rising and a falling phase—$y$ stimulates the growth of $Z$ at low concentrations and inhibits the growth of $Z$ at high concentrations, so the signal is toxic (negative growth rate) at both low and high levels of $y$.

As with the monophasic circuits, here there are also two possible cases. In the first case, the signal $y$ increases with tissue size $Z$ (i.e., $Z$ activates $y$). This circuit has a stable fixed point at $y = y_{ST}$ and an unstable fixed point at $y = y_{UST}$ where $y_{UST} < y_{ST}$ (Figs 1E and EV1C). In the second case, the signal $y$ decreases with tissue size ($Z$ inhibits $y$). This circuit also has a stable fixed point at $y = y_{ST}$ and an unstable fixed point at $y = y_{UST}$, but here $y_{UST} > y_{ST}$ (Figs 1F and EV1D).

In comparison with the monophasic circuits depicted in Fig 1A and B, the biphasic circuits have *fewer* types of sensing mutations with a fitness advantage. In particular, they are protected from invasion by loss-of-sensing mutants and locked-on sensing mutants. Whereas in the monophasic circuit of Fig 1A, loss-of-sensing mutations invade the population, in Fig 1E, the biphasic response gives these mutants a negative growth rate. They are thus eliminated. Similarly, whereas locked-on sensing mutants invade the monophasic circuit of Fig 1B, in the biphasic case of Fig 1F, they are eliminated (Fig EV2B). Thus, mutants with strong inactivation (or strong activation) in the response to $y$ have a fitness disadvantage (Fig 1H). This robustness to mutants is very important since such mutations may be common. For example, many constitutively active mutations of diverse G-coupled protein receptors have been observed (Seifert & Wenzel-Seifert, 2002), and it is common for mutations to lead to loss of function (Eyre-Walker & Keightley, 2007; Sarkisyan *et al*, 2016). Mutations with intermediate effects may be rarer; for example, a study in yeast (González *et al*, 2015) showed that mutations that destroy protein function are much more common than those that reduce its activity to an intermediate level.

The elimination of sensing mutants by the biphasic mechanism is frequency dependent: Mutants are eliminated if they have low frequency compared with wild-type cells. The reason for this is that when mutants are rare, the tissues maintain a proper signal $y_{ST}$ which the mutants mis-sense as $y_{MUT}$ and therefore have a fitness

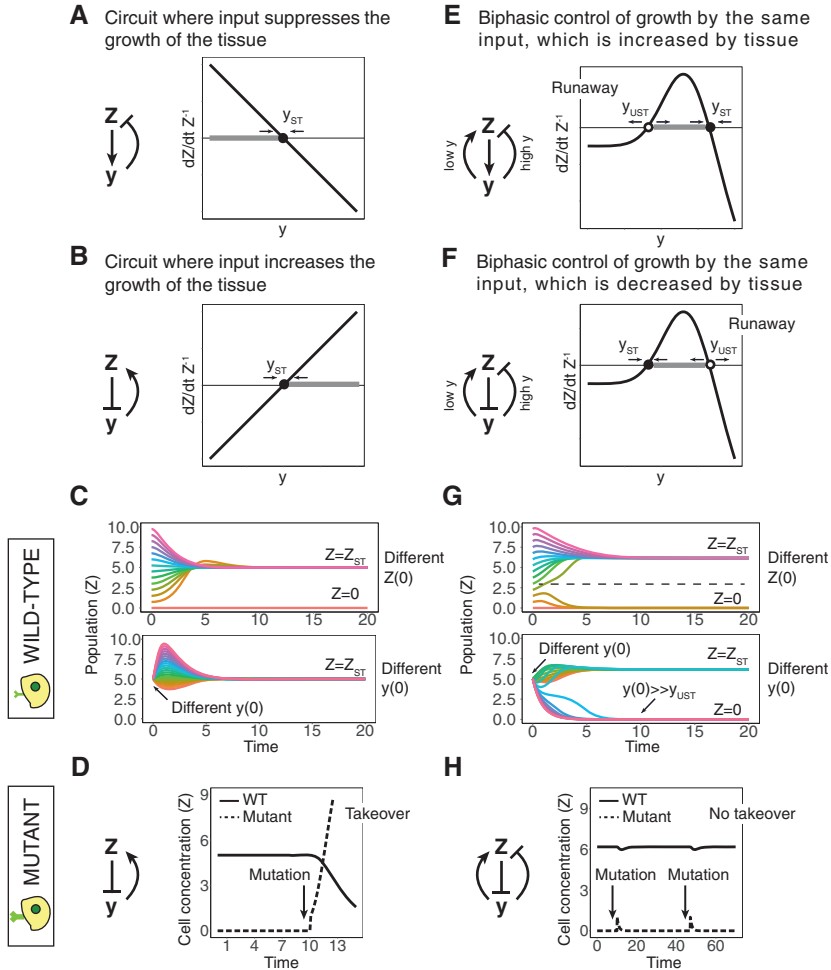

**Figure 1. Biphasic control can resist mutant invasion of feedback circuits.**

A  A monophasic feedback circuit in which cells $Z$ generate an input $y$ that inhibits their growth rate. The population is at steady state $Z = Z_{ST}$ when $y = y_{ST}$.

B  A monophasic feedback circuit where cells $Z$ decrease an input $y$, which increases their growth rate. The population is at steady state $Z = Z_{ST}$ when $y = y_{ST}$.

C  Trajectories of $Z$ from different initial concentrations of cells ($Z$) (i) or $y$ (ii) for the circuit of (B). The healthy concentration $Z = Z_{ST}$ is reached regardless of initial concentration of $Z$, as long as it is nonzero, and regardless of the initial concentration of $y$.

D  An arrow marks the time when a mutant with a strong activation of the sensing of $y$ arises (for the circuit depicted in B). This mutant has a selective advantage and takes over the population.

E  A biphasic feedback circuit where $Z$ generates a signal $y$, which, in turn, decreases the growth rate of $Z$ at high concentrations and increases the growth rate of $Z$ at low concentrations. The population is at steady state $Z = Z_{ST}$ when $y = y_{ST}$, and there is also an unstable fixed point at $y_{UST} < y_{ST}$.

F  A biphasic feedback circuit where cells $Z$ inhibit $y$, which, in turn, decreases the growth rate of $Z$ at high concentrations and increases the growth rate of $Z$ at low concentrations. The population is at steady state $Z = Z_{ST}$ when $y = y_{ST}$, and there is also an unstable fixed point at $y_{UST} > y_{ST}$.

G  Trajectories of $Z$ from different initial concentrations of $Z$ (i) or $y$ (ii) for the circuit depicted in (F). The healthy concentration $Z = Z_{ST}$ is not reached for small values of $Z$ ($Z << Z_{ST}$) or large values of $y$ ($y >> y_{UST}$).

H  The arrows mark the times when a mutant with a strong activation of the sensing of $y$ arises (for the biphasic circuit depicted in F). This mutant has a selective disadvantage and is thus eliminated.

disadvantage. On the other hand, if the mis-sensing mutant appears at high enough frequency, it is prevalent enough to change the level of $y$ and force it to reach an improper level that it mis-reads as $y_{ST}$. In this case, the population of mis-sensing mutants will be at steady state and will not be eliminated.

Biphasic circuits still have a range of mild-effect mutants with a growth advantage. These mutants mis-interpret the normal signal $y_{ST}$ as a different value lying in the gray-shaded regions of Fig 1E and F, namely $y_{MUT}$ lies between $y_{ST}$ and $y_{UST}$. Later, we discuss

mechanisms that can reduce the growth advantage of these mild mutants.

The biphasic mechanism of resistance to mutants, however, comes at a cost in terms of the robustness of the circuit to perturbations in the input $y$. The biphasic response curve crosses zero twice and therefore introduces a new *unstable* fixed point, denoted by a white circle in Fig 1E and F. The stable fixed point (full black circle) still exists, and the circuit can maintain the cell concentration constant in the face of small fluctuations around this fixed point.

However, large fluctuations in signal $y$ that exceed the unstable fixed point, or large fluctuations in $Z$, may lead to negative growth rate and to the risk of the elimination of the cell population (Fig 1G). Beyond the unstable fixed point, a runaway phenomenon occurs in which the cell population shrinks, leading to change in $y$ that pushes it deeper into the unstable region, leading to faster shrinkage and so on. This runaway phenomenon has the hallmarks of certain disease as described below.

To summarize so far, circuits with biphasic control avoid invasion by mutants with strong activation or inactivation of sensing. This robustness is useful because such mutations have a severe effect if they take over the population. This mechanism has two vulnerabilities: Mutations with mild effect on sensing may still invade, and an unstable fixed point introduced by the biphasic control provides risk of runaway behavior if signal fluctuates too widely. We next provide several examples of biphasic control.

## Glucotoxicity can protect from mutant beta cells, but can cause diabetes

The first example occurs in the endocrine circuit that regulates blood glucose by pancreatic beta cells (Fig 2). Fasting blood glucose ($y$) is maintained within a tight range around approximately $y_{ST}$ = 5 mM, and blood glucose dynamics are precise in response to perturbations (Allard *et al*, 2003; Ferrannini *et al*, 1985). To achieve this tight regulation, beta cells ($Z$) secrete insulin, which reduces glucose by increasing its uptake by peripheral tissues and decreasing its endogenous production. Thus, in this case, $Z$ inhibits $y$ (the case of Fig 1B and E).

The response of beta-cell growth to glucose is biphasic (Figs 2A and EV3A). Both low and high levels of glucose are toxic to beta cells. The response curve therefore has a stable fixed point at $y$ = 5 mM (Karin *et al*, 2016), and it also has an unstable fixed point at a higher glucose concentration. The toxicity at high levels of glucose is known as glucotoxicity (Efanova *et al*, 1998; Del Prato, 2009; Bensellam *et al*, 2012).

The unstable fixed point caused by glucotoxicity is puzzling, because, as described by Topp *et al* (2000), it provides a potential susceptibility to the system. If glucose levels exceed the unstable fixed point for extended periods (e.g., due to insulin resistance), beta cells will have negative growth rate and be removed, leading to an increase in glucose and a vicious cycle which can eliminate the beta-cell population. This process has been suggested to lead to type II diabetes (Topp *et al*, 2000; De Gaetano *et al*, 2008; Ha *et al*, 2016).

Glucotoxicity of beta cells is therefore detrimental, because it adds instability to perturbations in blood glucose. Since glucotoxicity is mediated by reactive oxygen species, it could have been mitigated by antioxidants, but antioxidants and oxidative stress protective genes are expressed at an exceptionally low level in beta cells (Robertson *et al*, 2003). This raises the question of why beta cells have not evolved to resist glucotoxicity. Here, we suggest that glucotoxicity may have a biological function: It eliminates beta-cell mutants with impaired glucose sensing.

Mutants that affect glucose sensing can occur at many steps in the glucose sensing and insulin secretion process. These steps include glucose import, phosphorylation and metabolism, closure of $K_{ATP}$ channels, and opening of voltage-dependent calcium channels (MacDonald *et al*, 2005). Many mutations are known to affect glucose sensing by beta cells (Fajans *et al*, 2001; James *et al*, 2009), among them autosomal dominant mutations in the enzyme glucokinase (GCK) (Froguel *et al*, 1993; Glaser *et al*, 1998; Fajans *et al*, 2001; Matschinsky, 2002; James *et al*, 2009).

We focus on GCK because it performs the rate-limiting step in glucose sensing. GCK is a hexokinase isozyme expressed in beta cells that phosphorylates the glucose that is transported into the cells. It has a half-maximal activation at $K$ = 8.4 mM glucose and a Hill coefficient of $n$ = 1.8 (Matschinsky, 2002). Hence, its activity level at the homeostatic set point of 5 mM glucose is ~30% of maximal activation. Glucose sensing is also affected by the expression level of GCK (Wang & Iynedjian, 1997). Germ line mutations in GCK result in low and high blood glucose levels for activating and inactivating mutations, respectively (Matschinsky, 2002). GCK mutations also affect the rates of beta-cell death and proliferation (Porat *et al*, 2011; Tornovsky-Babeay *et al*, 2014). This means that somatic mutations in GCK cause impaired glucose sensing that may alter circuit function.

Our theory makes a specific prediction on the fate of GCK mutations in beta cells, namely that their survival will be dependent on their frequency in the tissue (Fig 2B). We predict that a strong activating mutation in GCK will be lost when the majority of the population is wild type, because the mutants will be eliminated by the biphasic mechanism: Glucose level is set by the wild-type tissue to be 5 mM, but the mutants mis-sense a much higher level and succumb to glucotoxicity. In contrast, when the mutation is transmitted through the germ line, it is expected to survive and instead pull the blood glucose level to a low point, which the mutants interpret incorrectly as 5 mM.

This frequency-dependent survival was observed in an experiment by Tornovsky-Babeay *et al* (2014), which studied a strong (~6-fold) activating mutation of GCK (Y214C). This mutation, when transmitted via germ line, and thus to all beta cells, results in large and hyperfunctional islets and severe hypoglycemia (Cuesta-Munoz *et al*, 2004) as occurs in rare human patients. The experimenters conditionally expressed this mutation in beta cells of adult mice, such that only a subset of the beta cells expressed the transgene (~25%: 3 days after conditional expression). Both the proliferation and apoptosis rates increased in the cells expressing the transgene, but the increase in apoptosis rate was higher and after 22 days only 5% of the cells that expressed the transgene were left. Thus, the mutated cells were eliminated, whereas the wild-type cells remained.

We simulated this experiment using the biphasic control circuit (Fig 2B–D, see Appendix Section S3 for simulation details). The results of the simulation are consistent with the experimental observations—despite having a higher proliferation rate, the population of induced mutants was eliminated by their even higher apoptosis rate (Fig 2C). This elimination restores blood glucose levels to baseline after an initial hypoglycemia (Fig 2D), in agreement with the experimental blood glucose measurements (Fig 2C and D).

## Resistance to mutant invasion is enhanced by low proliferation, low cell number, and spatial compartments

We have thus seen that strong hypersensing mutants are eliminated from the beta-cell population. We now address the question of sensitivity to *mild* sensing mutants. These mutants misread the

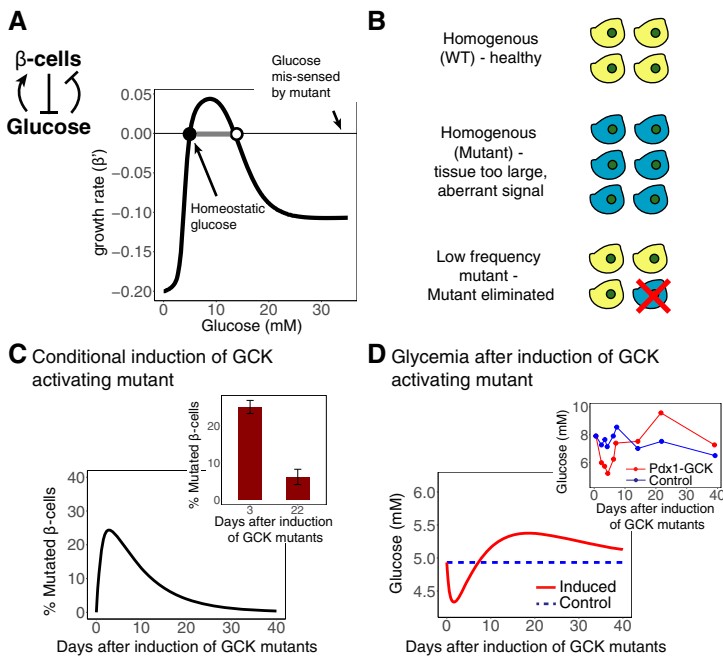

**Figure 2.  Frequency-dependent selection of mutant pancreatic beta cells.**

A     Feedback circuit in which beta cells secrete insulin, which lowers blood glucose levels. Blood glucose levels, in turn, affect beta-cell growth rate in a biphasic manner, with beta-cell growth being negative at both low and high glucose concentrations. The system is stable at the homeostatic glucose concentration at 5 mM. It also has an unstable fixed point at a higher glucose concentration. A mutant with a sixfold increase in glucokinase affinity senses the glucose level $y$ as $y_{MUT} = 6y$. This mis-sensing shifts its stable and unstable fixed points to lower glucose concentrations.

B     Biphasic control leads to frequency-dependent selection of the sensing mutant. The cell population in the tissue reaches a stable steady state only when it is homogenous with respect to the sensing of $y$. When a mutant with low frequency arises somatically, it is eliminated from the tissue; in contrast, if it is transmitted in the germ line, it will spawn a tissue with aberrant size.

C, D   Mathematical simulation of a tamoxifen-induced conditional knock-in of a sixfold activating GCK mutant in beta cells. (C) The percentage of beta cells with mutated GCK increases to ~25% after 3 days, but then decreases and is eliminated after a few weeks. (D) Glucose levels initially decrease after the tamoxifen injection, but return to normal after a few weeks. Insets: Experimental results of Tornovsky-Babeay *et al* (2014).

signal at a level that lies between the stable and unstable fixed points, $y_{ST} < y < y_{UST}$, leading to a growth advantage (shaded area of Fig 2A).

Using evolutionary dynamics theory (Nowak, 2006), we quantify the probability that such sensing mutants will invade the population of beta cells during a normal life span. The analysis results in several design principles to reduce the probability of such invasion.

To make simple approximation, we approximate the growth rate of beta cells when $y_{ST} < y < y_{UST}$ as constant, where $\lambda_+$ is the proliferation rate and $\lambda_-$ is the death rate in this range ($\lambda_+ > \lambda_-$). We also approximate by a constant the probability that a $k$-fold activating mutant will arise after a cell division, $\mu(k) = \mu_0$. The population of beta cells is sub-divided into compartments—pancreatic islets of Langerhans—each consisting of about $N \approx 3 \times 10^3 - 4 \times 10^3$ beta cells (Leslie & Robbins, 1995). We define the evolutionary stability of the circuit as the probability that no mutant will invade a single pancreatic islet by time $t$. This probability is given by a Moran process term (Appendix Sections S4 and S5):

$$\zeta(t) = e^{-N\tau^{-1}\delta\mu_0\left(1-\frac{1}{v}\right)t} \tag{1}$$

where $\delta = \frac{y_{UST} - y_{ST}}{y_{ST}}$ is the range of inputs to which the circuit is stable given by the relative distance between the stable and

unstable fixed points (dynamic stability of the circuit), $N$ is the number of beta cells in each islet, $\tau^{-1}$ is beta-cell turnover rate, and $v = \lambda_+/\lambda_-$ is the ratio of proliferation to removal for the mutants.

We can now approximate the evolutionary stability of the glucose homeostasis circuit using (equation 1). Typical parameters are $\delta \approx 1$ which corresponds to glucotoxicity around 10 mM glucose (Maedler *et al*, 2006) and $v \approx 3$ (Stolovich-Rain *et al*, 2012). For an average beta-cell turnover of $\tau^{-1} \approx 0.001$/day (Meier *et al*, 2008; Saisho *et al*, 2013) and mutation probability with a target size of 100 bp, $\mu_0 = 10^{-7}$, the evolutionary stability of the circuit is $\zeta(t) \approx e^{-2 \times 10^{-6}t}$. For a 70-year life span, the stability is $\zeta \approx 0.994$, so that ~0.6% of the pancreatic islets have an invading mutant by 70 years for these parameters. Note that we have only analyzed here mutations that are due to cell division and excluded other possible sources of somatic mutation, which may increase the overall number of islets with invading mutants. Increasing the mutation rate by 10-fold leads to ~6–7% of islets being taken over by an invading mutant.

The analysis shows that evolutionary stability is in a tradeoff with dynamic stability. The circuit can be made more dynamically stable by pushing the unstable fixed point to higher glucose (increasing δ). This, however, increases the range of mild mutants that

can invade (Fig 3A–C). At the extreme, one can push the unstable fixed point to infinity and end up with a monostable circuit, which is susceptible to all activating mutations in sensing.

Similarly, evolutionary stability is in tradeoff with the response time of the circuit. The response time to a glucose perturbation depends on the growth rate of the cells, which is, for $\tau^{-1} = \lambda_+$ : $\lambda_+ - \lambda_- = \tau^{-1} \cdot \left(1 - \frac{1}{v}\right)$. To make the response time more rapid, either $\tau^{-1}$ or $v$ should increase, but this will make the circuit less evolutionarily stable [decrease $\zeta$ according to (equation 1)]. The intuitive reason for this tradeoff is that fast response requires faster proliferation, but this gives mutants a bigger growth advantage. We conclude that if mutations that activate glucose sensing are sufficiently likely, glucotoxicity may be selected for despite its harmful potential for diabetes.

## Evolutionary stability of the parathyroid gland

Another prediction of (equation 1) is that changes in the parameters, such as a large increase in cell proliferation rate, can lead to invasion by mild sensing mutants. We hypothesize that this occurs in the circuit that controls calcium homeostasis, leading to the disease known as tertiary hyperparathyroidism.

The parathyroid (PT) gland ($Z$) controls plasma calcium ($y$) by secreting parathyroid hormone (PTH) which increases calcium production. This circuit is analogous to the glucose homeostasis circuit discussed previously—calcium controls both the secretion of

PTH and the mass dynamics of the PT gland (Naveh-Many *et al*, 1995; Wada *et al*, 1997; Mizobuchi *et al*, 2007). The signs of the circuit are opposite to the glucose circuit, because $Z$ acts to increase $y$. It is unclear whether this circuit has biphasic control, so the circuit is similar to either one of the circuits in Fig 1A or Fig 1E (Fig EV3B).

This circuit is sensitive to invasion by deactivating mutants in calcium sensing (mis-sensing calcium as lower than it actually is). Under normal conditions, however, the PT gland has a very low turnover (very small $\tau^{-1}$) (Bilezikian *et al*, 2001). Therefore, these mutants have a low probability to arise and invade due to the rarity of cell divisions.

However, in cases of increased demand for PTH, which occurs in hypocalcaemia such as that caused by renal failure, excessive proliferation of the parathyroid cells takes place ($\tau^{-1}$ increases). Such conditions are termed secondary hyperparathyroidism (SHPT). In such conditions, we expect, from (equation 1), that mutants will arise and have a large probability to invade the PT gland.

Indeed, the invasion of mutants with calcium-sensing inactivation often occurs in secondary hyperparathyroidism (Gogusev *et al*, 1997; Yano *et al*, 2000; Fraser, 2009). The invasion of such mutants alters the calcium homeostatic set point (Malberti, 1999) and leads to tertiary hyperparathyroidism. The new set point is mildly higher calcium, which is due to the mis-sensing of the mutants. The common mutations that lead to tertiary hyperparathyroidism are known to cause only an intermediate reduction in the expression of

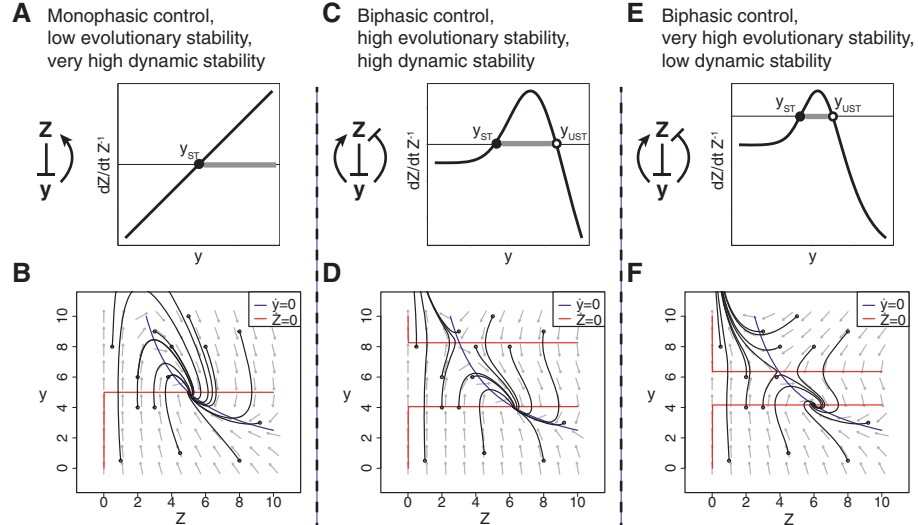

**Figure 3.  Tradeoff between evolutionary stability and dynamical stability.**

A   A monophasic feedback circuit where cells $Z$ inhibit $y$ which increases their growth rate. This circuit has low evolutionary stability—any mutant that mis-senses $y$ to a higher value may invade the population

B   For every initial value of $Z > 0$ or $y$, the circuit converges to the homeostatic set point. Nullclines are indicated by red and blue lines in the phase plots.

C   A biphasic feedback circuit where $Z$ inhibits $y$, which, in turn, decreases the growth rate of $Z$ at high concentrations and increases the growth rate of $Z$ at low concentrations. This circuit has high dynamical stability (large $y_{UST}$-$y_{ST}$) and high evolutionary stability, but mild activating mutants may invade the population.

D   Large perturbations in either $Z$ or $y$ may result in the elimination of the cell population $Z$ due to a runaway process. Nullclines are indicated by red and blue lines in the phase plots.

E   A biphasic feedback circuit with lower dynamical stability (small $y_{UST}$-$y_{ST}$) and higher evolutionary stability, since only few mild activating mutants may invade the population (gray region).

F   Small perturbations in either $Z$ or $y$ may result in the elimination of the cell population $Z$ due to a runaway process. Nullclines are indicated by red and blue lines in the phase plots.

                    

the calcium-sensing receptor (Yano *et al*, 2000) and not a strong inactivation. This is what we expect if calcium controls PT gland growth in a biphasic manner, eliminating the strong deactivating mutations.

## Biphasic control in secrete-and-sense circuits in T cells and bacteria

As an additional example, we consider the evolutionary stability of a motif suggested to control cell populations known as secrete and sense (You *et al*, 2004). An experimental characterization of such a circuit *in vitro* employed the control of T-cell population size by IL2, a cytokine secreted by the T cells (Hart *et al*, 2014). In this circuit, $y = IL2$ increases both the death rate and proliferation rate of the cells at different rates, similar to the circuit depicted in Fig 1E. The resulting overall growth rate is biphasic, with negative growth rate at low ($y < y_{UST}$) and high ($y > y_{ST}$) concentrations of *IL2* (Fig EV3C). This causes the population to have a stable fixed point at $y = y_{ST}$ and an unstable fixed point at $y = y_{UST}$. Initial seeding of T cells in plates across 4 decades of concentration led to convergence after 7 days to the same steady-state population to within a factor of 2 (Hart *et al*, 2014). This steady-state population was much lower than the carrying capacity of the system and resulted from vigorous balance of cell proliferation and death. Seeding with too few T cells, or experimental reduction in IL2, led to the elimination of the T cells. The present analysis suggests that the biphasic effects of IL2 can protect against loss-of-sensing mutants in IL2 signaling. This suggests an experiment in which such mutants are predicted to be eliminated if present at low concentrations within a wild-type population, but to take over if present at high numbers (frequency-dependent selection).

A secrete-and-sense circuit has also been synthetically engineered in bacteria by You *et al* (2004), by placing a death gene under control of a quorum sensing signal so the gene is activated when quorum signal is strong, similar to the circuit depicted in Fig 1A (Fig EV3D). This circuit maintains cell concentration constant. However, homeostatic control is rapidly lost (Balagadde, 2005) since selection favors mutants which inactivate the synthetic signaling pathway, in accord with the present predictions.

## Evolutionary stable strategies in tissues with stem cells

The cases discussed so far have a population of dividing cells that is under size control. Many tissues, however, are made of nondividing differentiated cells that originate from a pool of stem cells. We consider the case of tissues in which the differentiated cells are constantly removed and must be replenished, such as blood cells and the epithelia of lungs and skin. In these cases, tissue size control requires feedback from the differentiated cells back to the stem cells (Bullough, 1975). The dividing stem-cell population is sensitive to takeover by mutants. Here, we suggest that a biphasic control mechanism can provide protection from invasion of mutants also in this case (Fig 4).

We demonstrate the effect of biphasic control by considering a monophasic circuit presented by Buzi *et al* (2015). In this circuit, differentiated cells secrete a molecule *y* that affects the differentiation probability of the stem cells (Fig 4A and B). The molecule *y* increases the differentiation rate of the stem cells and thus limits their expansion rate. This type of feedback has been demonstrated

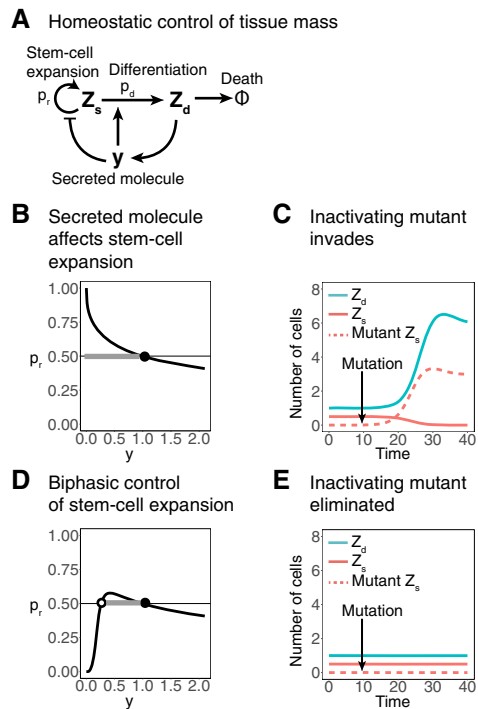

**Figure 4. Biphasic control can provide mutant resistance to stem-cell homeostatic circuits.**

A Homeostatic control of a population of cells $Z_d$, which differentiate from a population of dividing stem cells $Z_s$. Differentiated cells secrete a factor *y* which increases the differentiation rate $p_d$ of $Z_s$ and therefore decreases the rate of stem-cell expansion $p_r = 1 - p_d$.
B In a monophasic model, stem-cell expansion rate decreases with *y*. The system has a stable fixed point at the concentration of *y* where $p_r = 0.5$.
C A mutated stem cell with a strong inactivation of the sensing of *y* has a growth advantage (differentiates less), and therefore, it invades the stem-cell population. As a result, both the stem-cell pool and the number of terminally differentiated cells increase.
D Biphasic control of stem-cell expansion, where stem-cell expansion is low both at high and low concentrations of *y*. The system has a stable fixed point at the concentration of *y* where $p_r = 0.5$ and an unstable fixed point at some lower concentration of *y*.
E A mutated stem cell with a strong inactivation on *y* sensing now has a growth disadvantage and is therefore eliminated from the stem-cell population.

for many tissues, such as blood, skin, skeletal muscle, olfactory epithelium, bone, hair, and more (Lander *et al*, 2009; Buzi *et al*, 2015). In many of these tissues, the secreted molecule belongs to the TGF-β family (Lander *et al*, 2009). The dynamic equations for the stem cells $Z_s$ and differentiated cells $Z_d$ are as follows:

$$\dot{Z}_s = (2p_r(y) - 1)\lambda_+ Z_s \tag{2}$$

$$\dot{Z}_d = 2p_d(y)\lambda_+ Z_s - \lambda_- Z_d \tag{3}$$

$$y \propto Z_d \tag{4}$$

where $\lambda_+$ is the stem-cell division rate, $\lambda_-$ is the differentiated cell removal rate, $p_r$ is the probability that a stem cell that divided will not differentiate, and $p_d$ is the probability that it will ($p_r = 1 - p_d$). The differentiated cells secrete molecule *y* that increases

differentiation rate $p_d(y)$—forming a negative feedback loop (because differentiation is akin to loss of stem cells, leading to less differentiated tissue in the long term). Too many differentiated cells $Z_d$ lead to a high level of $y$ and to a decrease in the stem-cell population, leading to a reduction back to tissue size set point $Z_{dST}$. This monophasic circuit thus maintains a stable, constant population of differentiated cells (Buzi *et al*, 2015).

As above, this monophasic circuit is susceptible to invasion by loss-of-sensing mutations: stem cells that cannot sense $y$ or that mis-sense $y$ as too low. Within a single compartment, such a sensing mutant cannot coexist with wild-type stem cells. As this mutant stem cell differentiates less than the wild-type stem cells, it self-renews more often and has an evolutionary advantage over other stem cells. It is likely to invade the compartment and disrupt tissue homeostasis (Fig 4C). Invasion of the mutant means an exponential growth in both (mutant) stem cell and differentiated cell populations.

We find that adding a biphasic response curve can increase the evolutionary stability of this circuit (Fig 4D and E). In such biphasic control, $y$ stimulates the growth of the stem cells at low concentrations and also stimulates differentiation at high concentrations (and thus inhibits renewal at high concentrations). Therefore, stem cells with a strong inactivating mutants that mis-sense $y$ as too low grow less than wild type and thus have a selective disadvantage relative to other stem cells and are eliminated from the stem-cell population. The TGF-β feedback has indeed been demonstrated to have biphasic control in several cell types (Battegay *et al*, 1990; McAnulty *et al*, 1997; Cordeiro *et al*, 2000; Fosslien, 2009).

### Neuronal excitotoxicity as an additional putative case for evolutionary stability/disease tradeoff

Several other diseases are associated with a biphasic response of cells to their input. Glutamate, a common neurotransmitter, has a biphasic effect on neurons—it increases neuronal survival at intermediate concentrations and causes neuronal death at low and high levels (Lipton & Nakanishi, 1999; Fig EV3E). The latter effect is called neuronal excitotoxicity. Excitotoxicity is associated with neurodegenerative diseases such as Alzheimer's, Parkinson's, and Huntington's (Coyle & Puttfarcken, 1993; Dong *et al*, 2009).

We speculate that the biphasic effect of glutamate on neuronal survival may be beneficial for the elimination of neurons with improper sensing. Such defective neurons may arise due to somatic mutation in the brain, either in mature neurons (Lodato *et al*, 2015) or in neuronal progenitors (Poduri *et al*, 2013). In order to evaluate the role of the biphasic effect of glutamate on neuronal evolutionary stability, it is necessary to better characterize the homeostatic feedback circuits that control neuronal mass dynamics.

## Discussion

In this study, we raise the question of the stability of circuits that control tissue size with respect to invasion by mutants. We consider feedback circuits that provide size control by regulating cell growth according to an input signal proportional to the number of cells. We show that such feedback mechanisms can be invaded by commonly occurring mutants, which have loss-of-sensing or locked-on sensing of the input signal. Invasion leads to aberrant tissue size and function.

We find that these common mutants can be eliminated by a biphasic control mechanism. In biphasic control, the signal is toxic at both high and low levels, giving the mutants a selective disadvantage. The biphasic protection mechanism comes at a cost: It introduces an unstable fixed point that can cause a runaway phenomenon under strong fluctuations in the input signal, potentially leading to disease. This study thus provides an explanation for several well-studied toxicity phenomena associated with diseases, by suggesting that they have a beneficial function of protecting tissues from invasion by common mutants.

The biphasic control mechanism protects against strong sensing mutations, such as loss-of-function or locked-on receptors. These strong mutations presumably have a large mutational target size and are thus the most commonly arising mutations in a tissue. The control mechanism is sensitive, however, to a range of mild sensing mutations. These mutations cause the cell to mis-interpret the signal level, to a level that lies between the desired steady-state level $y_{ST}$ and the unstable fixed point $y_{UST}$ (Fig 1). It is likely that such mild mutations are more rare than loss-of-function or locked-on mutations. The vulnerability to these mild mutations might explain the recurrence of a few specific point mutations of mild effect in sensing pathways in cancer, presumably because mutations of larger effect are eliminated (Hanahan & Weinberg, 2011).

There is a tradeoff between evolutionary stability—the range of mild mutations that can invade, and dynamic stability—the position of the unstable fixed point. The closer the $y_{ST}$ is to $y_{UST}$, the higher the evolutionary stability and the lower the dynamical stability. As $y_{UST}$ approaches $y_{ST}$, we expect to see critical slowing down of the dynamics of the system and a general loss of resilience to perturbations (Scheffer *et al*, 2009). Such critical slowing down was shown to occur in populations of yeast in response to dilution (Dai *et al*, 2012) as well as in genetic circuits (Axelrod *et al*, 2015).

This study maps the concept of evolutionary stable strategies (ESS) from evolutionary ecology (Smith & Price, 1973) to the level of cell circuits in tissues. In ecology, an ESS is defined when a population of organisms with that strategy cannot be invaded by any other strategy. In the present study, a mis-sensing mutant is analogous to the invading strategy. In this sense, the biphasic mechanism is evolutionarily stable with respect to strong mutations. It is unstable to a range of mild mutations. The evolutionary instability to mild mutations can be reduced using compartments with small cell numbers, low turnover rates, and proximity of the stable and unstable fixed points, as described by equation (1).

As in ESS in ecology, selection of sensing mutants is frequency dependent: If the entire population is mutant, it can survive. But a single-mutant cell on a background of wild-type cells is eliminated by the biphasic mechanism. Experiments show the predicted frequency-dependent selection of a strong glucokinase mutant in beta cells: When present at low frequency, the mutants are eliminated; when present in the germ line, they survive and cause hypoglycemia.

Many of the biphasic toxicity phenomena considered here are mediated by excess production of reactive oxygen species (ROS) which leads to apoptosis (Coyle & Puttfarcken, 1993; Hildeman *et al*, 1999; Schulz *et al*, 2002; Robertson, 2004). ROS are implicated in both beta-cell glucotoxicity and neuronal excitotoxicity. Such toxicity can be mitigated by antioxidants, which reduce ROS levels (Skulachev, 1998). Thus, the level of antioxidants may, in principle, tune the tradeoff between evolutionary stability and dynamic stability

described here. High antioxidants can reduce the toxicity of high signal level and thus push the unstable fixed point farther from the stable fixed point. This can reduce the risk of disease, but increase susceptibility to invasion by mild mutants. This tradeoff may provide a viewpoint to understand the conflicting effects of antioxidants on health (Bjelakovic *et al*, 2012; Sayin *et al*, 2014; Le Gal *et al*, 2015).

In this study, we discussed circuits where a tissue regulates its own size. Some tissues, however, regulate the size of other tissues. For example, the ovaries regulate mammary epithelial mass by secreting estrogen, and the pituitary gland regulates the mass of the thyroid and adrenal glands by secreting TSH and ACTH, respectively. Depending on the feedback loops at play, such circuits may be susceptible to mutant invasion both in the regulating and regulated tissue. The considerations of this study indicate that biphasic control reduces the susceptibility to invading mutants in these cases as well. We therefore predict biphasic responses also when tissues regulate each other. For example, estrogen controls mammary growth in a biphasic manner (Lewis-Wambi & Jordan, 2009), therefore reducing the target range of mutants with a fitness advantage in the mammary epithelium.

Finally, biphasic control raises the question of how tissues can start growing. Consider the tissue in Fig 1E, in which $Z$ produces $y$. If initially $y = 0$, $Z = \varepsilon$, then the tissue has negative growth rate and cannot grow to reach $Z = Z_{ST}$. This can be resolved if $y$ is determined externally during tissue development. For example, during gestation, metabolites and factors are supplied to the fetus externally by the mother at levels close to $y_{ST}$. Another possibility is that tissue development is determined by a different program that is later suppressed.

In summary, we show that physiological feedback circuits are inherently vulnerable to takeover by mutants that mis-sense the feedback signal. Biphasic mechanisms, in which the signal is toxic at both high and low levels to the relevant tissue, can protect against such mutant invasion. We therefore hypothesize that phenomena such as glucotoxicity and excitotoxicity may reflect the bad side of a good anti-mutant strategy (Stearns & Medzhitov, 2016). Characterizing physiological homeostatic circuits and the tradeoffs they face in quantitative detail may thus lead to a better understanding of diabetes (Topp *et al*, 2000), neurodegenerative diseases (Doble, 1999; Lipton & Nakanishi, 1999), and possibly other pathologies associated with biphasic control.

## Materials and Methods

### Circuits with monophasic and biphasic control

To simulate the circuits of Fig 1 in the main text, we used a circuit where a cell mass $Z$ either increases the level of its input $y$ (Fig 1A and E) or decreases the level of $y$ (Fig 1B and F). The equation used for $Z$ is follows:

$$\dot{Z} = Z \cdot (\lambda_+(y) - \lambda_-(y)) = Z \cdot \lambda(y) \tag{5}$$

where $\lambda_+$ is the $y$-dependent proliferation rate of $Z$, and $\lambda_-$ is the $y$-dependent removal rate of $Z$.

In Fig 1C, D, G and H, we simulated two cases—a monophasic circuit, where $y$ increases the growth rate of $Z$, and a biphasic

circuit, where $y$ increases the growth rate of $Z$ at low concentrations and decreases the growth rate of $Z$ at high concentrations. The monophasic circuit was simulated using the growth rate equations:

$$\lambda_+(y) = \frac{y}{10} \tag{6}$$

$$\lambda_-(y) = 0.5 \tag{7}$$

and the biphasic circuit was simulated by using the growth rate equations:

$$\lambda_+(y) = \frac{4.8}{1 + \left(\frac{7}{y}\right)^5} \tag{8}$$

$$\lambda_-(y) = \frac{6}{1 + \left(\frac{8}{y}\right)^5} + 0.1 \tag{9}$$

These circuits were also used to simulate the phase plots in Fig 3A and B. For Fig 3C, we used the following circuit:

$$\lambda_+(y) = \frac{4.8}{1 + \left(\frac{5.5}{y}\right)^6} \tag{10}$$

$$\lambda_-(y) = \frac{6}{1 + \left(\frac{6.3}{y}\right)^6} + 0.3 \tag{11}$$

We used the following equation for the dependence of $y$ on $Z$:

$$\dot{y} = \mu \cdot (M - Zy) \tag{12}$$

This equation means that $Z$ increases the degradation rate of $y$, and at steady state, we get $Z_{st}y_{st} = M$. We chose the parameters $M = 25, \mu = 0.25$.

### Mutant invasion simulation

We simulated the effect of a mutation by adding a term $Z_{mut}$ such that:

$$\dot{y} = \mu \cdot (M - (Z + Z_{mut})y) \tag{13}$$

$Z_{mut}$ represents the mass of cells with a (given) $k$-fold sensing mutation on $y$, so the growth rate of $Z_{mut}$ is given as follows:

$$\dot{Z}_{mut} = Z_{mut}\lambda(ky) = Z_{mut}(\lambda_+(ky) - \lambda_-(ky)) \tag{14}$$

Note that for the monophasic circuit simulated in Fig 1, the removal rate $\lambda_-$ does not depend on $y$, and therefore, it is not affected by the sensing mutation (only $\lambda_+$ is affected). We simulated the invasion of a fourfold sensing mutant in Fig 1D and H by setting $Z_{mut} \leftarrow 1$ at specific time intervals in the simulation ($t = 10$ for the monophasic circuit and $t = 10$, $t = 47$ for the biphasic circuit). The initial values for the simulations were $Z_{mut0} \leftarrow 0$, $Z_0 \leftarrow 5$, $y_0 \leftarrow 4$ for the monophasic circuit in Fig 1D, and $Z_{mut0} \leftarrow 0$, $Z_0 \leftarrow 6.16$, $y_0 \leftarrow 4.06$ for the biphasic circuit in Fig 1H.

## Circuits of communicating stem cells

In this study, we presented two circuits that regulate the functional mass of differentiated cells, based on the model that is presented in Buzi *et al* (2015). For the monophasic circuit, the equations are as follows:

$$\dot{Z}_s = (2p_r(y) - 1)\lambda_+ Z_s \tag{15}$$

$$\dot{Z}_{s_{mut}} = (2p_r(ky) - 1)\lambda_+ Z_{s_{mut}} \tag{16}$$

$$\dot{Z}_d = 2(1 - p_r(y))\lambda_+ Z_s + 2(1 - p_r(ky))\lambda_+ Z_{smut} - \lambda_- Z_d \tag{17}$$

$$y \propto Z_d \tag{18}$$

where $\lambda_+$ is the stem-cell division rate, $\lambda_-$ is the differentiated cell removal rate, $p_r$ is the probability that a stem cell that divided will not differentiate, and $1 - p_r$ is the probability that it will differentiate. The population $Z_{s_{mut}}$ is the population of stem cells with a k-fold sensing mutation. The monophasic replication rate $p_r(y)$, which is depicted in Fig 3B, was set as follows:

$$p_r(y) = \frac{1}{1 + \sqrt{y}} \tag{19}$$

The exact function used is not important, since as long as it is monotonically decreasing, an invading mutant will take over. In the biphasic case, the replication rate used is as follows:

$$p_r(y) = \frac{1}{1 + \sqrt{y}} \cdot \frac{1}{1 + \left(\frac{1}{5y}\right)^4} \tag{20}$$

The simulation of invading mutants is the same as for Fig 1 (which is explained in the mutant invasion simulation section). For the simulations, we set $\lambda_+ \leftarrow 1, \lambda_- \leftarrow 0.5$, $k = \frac{1}{6}$, and with the initial conditions $Z_{s0} \leftarrow 0.5$, $Z_{s_{mut}0} \leftarrow 0$, $Z_{d0} \leftarrow 1$. A mutation event was set such that $Z_{s_{mut}} \leftarrow 0.01$ at $t = 10$.

Expanded View for this article is available online.

## Acknowledgements
This work was supported by the Israel Science Foundation (1349/15) and the Minerva Foundation. UA is the incumbent of the Abisch-Frenkel Professorial Chair. OK is supported by the Azrieli Center for Systems Biology grant.

## Author contributions
OK and UA conceived and performed the research. OK and UA wrote the manuscript.

## Conflict of interest
The authors declare that they have no conflict of interest.

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
