## [Review Process File · Molecular Systems Biology]

Biphasic response as a mechanism against mutant takeover in tissue homeostasis circuits

Uri Alon & Omer Karin

Corresponding author: Omer Karin, Weizmann Institute of Science

Review timeline:

Submission date:	24 February 2017
Editorial Decision:	24 March 2017
Revision received:	09 April 2017
Editorial Decision:	11 May 2017
Revision received:	15 May 2017
Accepted:	22 May 2017

Editor: Maria Polychronidou

Transaction Report:

1st Editorial Decision

24 March 2017

Thank you again for submitting your work to Molecular Systems Biology. We have now heard back from the two referees who agreed to evaluate your manuscript. As you will see below, the reviewers acknowledge that the study seems very interesting. They raise however a series of concerns, most of which can be addressed by text modifications, which we would ask you to address in a revision of the manuscript.

I think that the recommendations of the reviewers are quite clear so there is no need to repeat any of the points listed below. Please feel free to contact me in case you would like to discuss any of the points in further detail.

REVIEWER REPORTS

Reviewer #1:

This is a very interesting manuscript that provides a deep and novel understanding of how tissues maintain their own size and how robust that maintenance is. Many tissues seem to sense and control their own size. This makes them vulnerable to mutations that affect size sensing, such that the tissue perceives itself smaller than it actually is. Tissues can protect themselves from such mutational events by a "biphasic response" where large-effect mutations have negative fitness (causing mutants to die and disappear from the population). However, this mechanism has two drawbacks: (i) small-effect mutations can still invade and (ii) large changes in the signal level can create a positive

feedback of cell death that leads to the extinction of the cell population. The applications of the theory are discussed in several test cases, including an explanation of glucotoxicity and diabetes, the evolutionary stability of the parathyroid gland, loss of T-cells, and tissue size stability based on stem cells.

Considering the depth of understanding gained, the interdisciplinary approach used and the high potential for broad interest, the manuscript is a great candidate for publication in *Molecular Systems Biology*. I only have some minor comments on data presentation and references.

- (1) Generally, the feedback could act with a delay. What is known about the delay for various systems, and how would it affect the conclusions? This would be worth discussing.
- (2) In some cases, a signal from one tissue controls the size of another. For example, estrogen produced mainly in the ovaries controls breast epithelial tissue size. How could such tissues control their own size if they are controlled by signal from another tissue? This may be worth discussing.
- (3) The manuscript studies chemical feedback for tissue size maintenance. It may also be worth briefly discussing the sensing of mechanical signals and mechanical feedback as an additional way of tissue size control. Could mechanical feedback ever be biphasic?
- (4) Is there a way to graphically illustrate the effects of mutants on at least one of the plots in Fig. 1A-F? Possibly as a shift in the response curve?
- (5) Fig. 2D, inset. It seems like the colors are reversed compared to the theoretical graph. For clarity, the same color scheme would be best to use (control should be red in both plots).
- (6) Fig. 4E: The effect of mutations in the time course is really invisible. Is there a way to choose parameters that make these effects easier to see? It would help the reader.
- (7) It may be worth adding a new figure, with separate panels illustrating as a cartoon diagram each example system (glucotoxicity, etc.) discussed in the second half of the manuscript. This comment excludes stem cell based homeostasis, for which a diagram exists in Fig. 4.
- (8) The biphasic response protects from mutations because most of them are loss-of-function mutations: "it is common for mutations to lead to loss of function". On the other hand, mutations with intermediate effects are relatively rare. A recent study [Gonzalez et al., *Mol. Syst. Biol.* 11(8):827 (2015), PMID=26324468] may be worth citing since it provides evidence for this in yeast, showing that many mutations can destroy protein function, while only a few mutations can fine-tune the protein function to satisfy two opposing requirements.
- (9) Fig. 1C in the recent paper by Karin et al. (2016) has additional regulatory links that ensure dynamical compensation. How would adding these links affect the conclusions of this manuscript?
- (10) It may be worth discussing the effects of modeling assumptions and doing some parameter scans. How robust are the conclusions to parameter choices and other modeling assumptions? Is there a difference between assuming exponential versus logistic growth for tissues in the models?

Reviewer #2:

In this paper, Karin and Alon explore how a biphasic response allows for improved non-invasibility of tissue homeostasis circuits. I have not heard of this argument before, and I found it quite elegant. I thought that the paper was well written and the science both interesting and convincing. I therefore recommend publication in *MSB* with the minor changes listed below.

My major concern had to do with the presentation in the text of Figure 1, where it was not very clear that biphasic response is expected to yield two qualitatively different responses depending upon where the negative regulation is located (Fig 1E vs 1F). The distinction of these two cases was nice and clear for Fig 1A and 1B, but the distinction was glossed over for 1E and 1F, where the difference is more interesting.

More generally, the dynamics of these models should be specified in the full Z-y plane (as illustrated later in Fig 3). It may be nice to include a supplementary figure with the phase portrait of how the system evolves dynamically over all (Z,y) for these different models. I am used to thinking about $(1/Z)dZ/dt$ as a function of Z (instead of as a function of y).

I find the idea of a biphasic response to be quite interesting, but it does lead to the question of how the tissue can start growing, since initially $y < y_{UST}$ (Fig 1E). This might be something to discuss somewhere.

I very much liked the idea that there is a trade-off between evolutionary stability and dynamic stability (as well as response time, which is actually the same thing). It may be interesting for the authors to note that this is a general property, since as the unstable fixed point approaches the stable fixed point there will be critical slowing down in which the dominant eigenvalue goes to zero. There will also in general be a loss of resilience to perturbations as these fixed points approach each other. We observed this in both populations (Dai et al, Science (2012)) as well as in gene circuits (Axelrod et al, eLife (2015)).

The authors might want to work harder to explain why there is frequency dependence in these models, since this is a subtle point and it was not always clear to me when frequency dependence would be present.

Minor comments:

line 44: "invade" does not require a take-over, but simply spreading when rare.

line 105: It may be helpful to point out that after this mutant fixes the resulting tissue will reach a new equilibrium size that is larger than the previous one. In addition, this new tissue will be susceptible to invasion of a new mutant with a larger equilibrium tissue size, etc. I leave this up to the authors.

I enjoyed reading this paper. Thanks.

1st Revision - authors' response

09 April 2017

Thank you very much for the positive consideration of our manuscript and for the reviewer comments. We have now addressed all of the comments in the revised manuscript. We detail below the point-by-point changes.

Reviewer #1:

This is a very interesting manuscript that provides a deep and novel understanding of how tissues maintain their own size and how robust that maintenance is. Many tissues seem to sense and control their own size. This makes them vulnerable to mutations that affect size sensing, such that the tissue perceives itself smaller than it actually is. Tissues can protect themselves from such mutational events by a "biphasic response" where large-effect mutations have negative fitness (causing mutants to die and disappear from the population). However, this mechanism has two drawbacks: (i) small-effect mutations can still invade and (ii) large changes in the signal level can create a positive feedback of cell death that leads to the extinction of the cell population. The applications of the theory are discussed in several test cases, including an explanation of glucotoxicity and diabetes, the evolutionary stability of the parathyroid gland, loss of T-cells, and tissue size stability based on stem cells.

Considering the depth of understanding gained, the interdisciplinary approach used and the high potential for broad interest, the manuscript is a great candidate for publication in Molecular Systems Biology. I only have some minor comments on data presentation and references.

We thank the reviewer for this endorsement.

(1) Generally, the feedback could act with a delay. What is known about the delay for various systems, and how would it affect the conclusions? This would be worth discussing.

We thank the reviewer for this comment. In the revised supplementary information we added a new section (Supplementary Information 2) that tests whether adding a delay term affects the resistance to invasion of the circuits that were described in Figure 1. We also explicitly modelled glucose dynamics with differential equations for insulin, glucose (see point 9). The new section is as follows:

Supplementary Figure 2 Simulation of an event where a strong activating mutant arises either in a circuit with monophasic control (A-C) or biphasic control (D-F). The arrows mark the times when a mutant with a strong activation of the sensing of y arises. The circuits are similar to the circuits depicted in Fig. 1B and Fig. 1F, except that Z acts on y with delay modeled by an intermediate variable r with delay parameter τ . As was the case without r , also here the monophasic circuit is susceptible to mutant invasion whereas the biphasic circuit is not.

In the main text, we analyzed circuits where cells Z adjust their own growth rate as a function of a signal y , which, in turn, is affected by the size of the tissue. Here, we consider the case where y affects Z with a delay. Delays occur in endocrine circuits, where the level of the regulated variable (e.g. blood glucose) is controlled with a delay relative to its regulating hormone (insulin).

In the examples of Figure 1 we used the following equations to model the mutant resistance of the circuits in Fig. 1BF:

$$\dot{y} = \mu \cdot (M - (Z + Z_{mut})y) \quad [1]$$

$$\dot{Z} = Z \cdot (\lambda_+(y) - \lambda_-(y)) \quad [2]$$

We tested whether adding a delay to this system affects the resistance of monophasic or biphasic circuits to sensing mutants. To do so, we modify the equations so they include an intermediate variable r with a typical timescale τ :

$$\dot{r} = \tau \cdot (Z + Z_{mut} - r) \quad [1]$$

$$\dot{y} = \mu \cdot (M - ry) \quad [2]$$

$$\dot{Z} = Z \cdot (\lambda_+(y) - \lambda_-(y)) \quad [3]$$

The parameter τ represents the delay of the system. We tested the effect of 3 different values of τ on the resistance to mutants (Supplementary Figure 2) - $\tau=0.01$ (slow), $\tau=1$ (intermediate) and $\tau=100$ (fast). For all these values of τ , an activating mutant invades the monophasic circuit but does not invade the biphasic circuit.

“

(2) In some cases, a signal from one tissue controls the size of another. For example, estrogen produced mainly in the ovaries controls breast epithelial tissue size. How could such tissues control their own size if they are controlled by signal from another tissue? This may be worth discussing.

We thank the reviewer for this comment. In the revised discussion, we have now added a new paragraph discussing circuits where one tissue controls the size of another tissue:

Page 19:

“

In this study we discussed circuits where a tissue regulates its own size. Some tissues, however, regulate the size of other tissues. For example, the ovaries regulate mammary epithelial mass by secreting estrogen, and the pituitary gland regulates the mass of the thyroid and adrenal glands by secreting TSH and ACTH respectively. Depending on the feedback loops at play, such circuits may be susceptible to mutant invasion both in the regulating and regulated tissue. The considerations of this study indicate that biphasic control reduces the susceptibility to invading mutants in these cases as well. We therefore predict biphasic responses also when tissues regulate each other. For example, estrogen controls mammary growth in a biphasic manner (Lewis-Wambi and Jordan, 2009), therefore reducing the target range of mutants with a fitness advantage in the mammary epithelium.

“

(3) The manuscript studies chemical feedback for tissue size maintenance. It may also be worth briefly discussing the sensing of mechanical signals and mechanical feedback as an additional way of tissue size control. Could mechanical feedback ever be biphasic?

We thank the reviewer for this comment. We now mention in the introduction an example in which mechanical feedback can also be biphasic:

Page 4:

“

Biphasic control of growth is prevalent in physiological systems. Examples include the control of beta cell mass by glucose (Robertson et al., 2003), the control of mammary gland mass by estrogen (Lewis-Wambi and Jordan, 2009), the control of neuronal survival by glutamate (Hardingham and Bading, 2003), epidermal growth factor signaling (Högnason et al., 2001) and the control of T-cell concentration by IL2 and by antigen level (Critchfield et al., 1994; Hart et al., 2014). **Biphasic control was also demonstrated for mechanical signaling - the control of epithelial cell proliferation by mechanical stretch through Piezo1 (Gudipaty et al., 2017).** In all of these cases, signal is toxic at both high and low levels.

“

(4) Is there a way to graphically illustrate the effects of mutants on at least one of the plots in Fig. 1A-F? Possibly as a shift in the response curve?

We added an expanded view figure (EV Fig. 2) that shows the effect of mutants on the response curve of cell growth:

(5) Fig. 2D, inset. It seems like the colors are reversed compared to the theoretical graph. For clarity, the same color scheme would be best to use (control should be red in both plots).

Fixed.

(6) Fig. 4E: The effect of mutations in the time course is really invisible. Is there a way to choose parameters that make these effects easier to see? It would help the reader.

Fixed.

(7) It may be worth adding a new figure, with separate panels illustrating as a cartoon diagram each example system (glucotoxicity, etc.) discussed in the second half of the manuscript. This comment excludes stem cell based homeostasis, for which a diagram exists in Fig. 4.

We added an expanded view figure (EV Fig. 3) with cartoon diagrams for the different circuits:

(8) *The biphasic response protects from mutations because most of them are loss-of-function mutations: "it is common for mutations to lead to loss of function". On the other hand, mutations with intermediate effects are relatively rare. A recent study [Gonzalez et al., Mol. Syst. Biol. 11(8):827 (2015), PMID=26324468] may be worth citing since it provides evidence for this in yeast, showing that many mutations can destroy protein function, while only a few mutations can fine-tune the protein function to satisfy two opposing requirements.*

We thank the reviewer for this comment. In the revised results, we now refer to this study:

Page 6.

“

Mutations with intermediate effects may be rarer; for example, a study in yeast (González et al., 2015) showed that mutations that destroy protein function are much more common than those that reduce its activity to an intermediate level.

“

(9) *Fig. 1C in the recent paper by Karin et al. (2016) has additional regulatory links that ensure dynamical compensation. How would adding these links affect the conclusions of this manuscript?*

We thank the reviewer for this comment. In the revised supplementary information we now added a figure that shows the results from Fig. 2 when we explicitly simulate glucose and insulin dynamics.

(10) *It may be worth discussing the effects of modeling assumptions and doing some parameter scans. How robust are the conclusions to parameter choices and other modeling assumptions? Is there a difference between assuming exponential versus logistic growth for tissues in the models?*

We thank the reviewer for this comment. In the revised supplementary information we now added a new section (Supplementary Information 1) that shows that the conclusions of the manuscript hold for both exponential and logistic growth. We also present there simulations of mutant invasion for monophasic and biphasic circuits, with different parameter values for the carrying capacity. The section is as follows:

“

Supplementary Figure 1. Adding carrying capacity K to the circuits preserves the conclusions of the study. Simulation of an event where a strong activating mutant arises either in a circuit with monophasic control (A-C) or biphasic control (D-F) with logistic growth with a carrying capacity K . The arrows mark the times when a mutant with a strong activation of the sensing of y arises. As was the case for exponential growth, also under logistic growth the monophasic circuit is susceptible to mutant invasion whereas the biphasic circuit is not.

In this section, we ask whether changing exponential growth to logistic growth in the circuits affects the conclusions. In the main text, we analyzed circuits where cells Z adjust their own growth rate as a function of a signal y , which, in turn, is affected by the size of the tissue. The signal y affects the growth rate of cells by affecting either their proliferation or removal rate, so we can model the dynamics of Z using the following equation:

$$\dot{Z} = Z \cdot (\lambda_+(y) - \lambda_-(y)) \quad [1]$$

Where λ_+ is the y -dependent proliferation rate of Z and λ_- is the y -dependent removal rate of Z . As discussed in the main text, the feedback on Z through y can robustly maintain tissue size, but is susceptible to the invasion of mis-sensing mutants.

The growth rate of Z can be either logistic or exponential. Exponential growth means that the production rate λ_+ does not depend on Z (for example $\lambda_+=y$), and is relevant when the cells are far from carrying capacity. When the cells are closer to carrying capacity, however, a logistic model more appropriately models the dynamics of Z :

$$\dot{Z} = Z \cdot \left(\lambda_+(y) \cdot \left(1 - \frac{Z}{K}\right) - \lambda_-(y) \right) \quad [1]$$

In which proliferation rate drops to zero as cells approach the carrying capacity K .

The conclusions of the manuscript hold both when the growth of the cells is logistic or exponential (Supplementary Figure 1): the biphasic circuit is resistant whereas the monophasic circuit is not.

Reviewer #2:

In this paper, Karin and Alon explore how a biphasic response allows for improved non-invasibility of tissue homeostasis circuits. I have not heard of this argument before, and I found it quite elegant. I thought that the paper was well written and the science both interesting and convincing. I therefore recommend publication in MSB with the minor changes listed below.

We thank the reviewer for this endorsement.

My major concern had to do with the presentation in the text of Figure 1, where it was not very clear that biphasic response is expected to yield two qualitatively different responses depending upon where the negative regulation is located (Fig 1E vs 1F). The distinction of these two cases was nice and clear for Fig 1A and 1B, but the distinction was glossed over for 1E and 1F, where the difference is more interesting.

We thank the reviewer for this comment. In the revised results, we now presented in the text the distinction between Fig. 1E and Fig. 1F:

Page 6:

“

As with the monophasic circuits, here there are also two possible cases. In the first case, the signal y increases with tissue size Z (that is, Z activates y). This circuit has a stable fixed point at $y=y_{ST}$ and an unstable fixed point at $y=y_{UST}$ where $y_{UST} < y_{ST}$ (Fig. 1E and EV Fig. 1C). In the second case, the signal y decreases with tissue size (Z inhibits y). This circuit also has a stable fixed point at $y=y_{ST}$ and an unstable fixed point at $y=y_{UST}$, but here $y_{UST} > y_{ST}$ (Fig. 1F and EV Fig. 1D).

“

More generally, the dynamics of these models should be specified in the full Z - y plane (as illustrated later in Fig 3). It may be nice to include a supplementary figure with the phase portrait of how the system evolves dynamically over all (Z,y) for these different models. I am used to thinking about $(1/Z)dZ/dt$ as a function of Z (instead of as a function of y).

We thank the reviewer for this comment. We added an expanded view figure (EV Fig. 1) with full specifications of the models from Figure 1 in the Z - y plane:

I find the idea of a biphasic response to be quite interesting, but it does lead to the question of how the tissue can start growing, since initially $y < y_{UST}$ (Fig 1E). This might be something to discuss somewhere.

We thank the reviewer for this comment. In the revised discussion, we now address this question in a new paragraph on page 19:

Page 19:

“

Finally, biphasic control raises the question of how tissues can start growing. Consider the tissue in Fig. 1E, in which Z produces y . If initially $y=0$, $Z=\epsilon$ then the tissue has negative growth rate and cannot grow to reach $Z=Z_{ST}$. This can be resolved if y is determined externally during tissue development. For example, during gestation, metabolites and factors are supplied to the fetus externally by the mother at levels close to y_{ST} . Another possibility is that tissue development is determined by a different program that is later suppressed.

“

I very much liked the idea that there is a trade-off between evolutionary stability and dynamic stability (as well as response time, which is actually the same thing). It may be interesting for the

authors to note that this is a general property, since as the unstable fixed point approaches the stable fixed point there will be critical slowing down in which the dominant eigenvalue goes to zero. There will also in general be a loss of resilience to perturbations as these fixed points approach each other. We observed this in both populations (Dai et al, Science (2012)) as well as in gene circuits (Axelrod et al, eLife (2015)).

We thank the reviewer for this comment. We addressed this point in the revised discussion.

Page 18:

“

There is a tradeoff between evolutionary stability – the range of mild mutations that can invade, and dynamic stability- the position of the unstable fixed point. The closer y_{ST} is to y_{UST} , the higher the evolutionary stability and the lower the dynamical stability. As y_{UST} approaches y_{ST} , we expect to see critical slowing down of the dynamics of the system and a general loss of resilience to perturbations (Scheffer et al., 2009). Such critical slowing down was shown to occur in populations of yeast in response to dilution (Dai et al., 2012) as well as in genetic circuits (Axelrod et al., 2015).

“

The authors might want to work harder to explain why there is frequency dependence in these models, since this is a subtle point and it was not always clear to me when frequency dependence would be present.

We thank the reviewer for this comment. We addressed this point in the revised results section.

Page 7.

“

The elimination of sensing mutants is frequency-dependent: mutants are eliminated if they have low frequency compared with wild-type cells. The reason for this is that when mutants are rare, the tissues maintains a proper signal y_{ST} which the mutants mis-sense as y_{MUT} , and therefore have a fitness disadvantage. On the other hand, if the mis-sensing mutant appears at high enough frequency, it is prevalent enough to change the level of y so that it mis-senses it as y_{ST} (although the true level of y will be higher or lower than y_{ST}). In this case, the population of mis-sensing mutants will be at steady-state and will not be eliminated.

“

Minor comments:

line 44: "invade" does not require a take-over, but simply spreading when rare.

Fixed.

line 105: It may be helpful to point out that after this mutant fixes the resulting tissue will reach a new equilibrium size that is larger than the previous one. In addition, this new tissue will be susceptible to invasion of a new mutant with a larger equilibrium tissue size, etc. I leave this up to the authors.

Fixed.

I enjoyed reading this paper. Thanks.

Thank you.

2nd Editorial Decision

11 May 2017

Thank you for submitting your revised study. We have now heard back from the referee who was asked to evaluate your manuscript. As you will see below, s/he is satisfied with the modifications made and supports publication of the study.

We recently implemented a model curation service for papers that contain mathematical models. This is done together with Prof. Jacky Snoep and the FAIRDOM team. In brief, the aim is to enhance reproducibility and add value to papers containing mathematical models. Jacky Snoep's summary on the model curation (*Model Curation Report*) is pasted below the reviewer's comments. here are some minor issues, which we would ask you to fix when you submit your revision.

 REVIEWER REPORT

Reviewer #2:

The authors have addressed my questions. I think that this is a beautiful paper, and will be of interest to the broad readership of MSB.

 MODEL CURATION REPORT

Technical curation for the mathematical models in MSB-17-7559R

The models described in the manuscript are clearly meant to be of a generic nature and not highly dependent on specific parameter values or initial conditions. However, to reproduce the figures in the manuscript it is necessary to have a full description of the model and it is MSB policy to have a full model description either in the manuscript or in supplementary material.

Below I give a summary of the e-mail communication with the authors, to clarify the model description in the manuscript.

With the additional information given by the authors, the model simulations given in the manuscript could be reproduced.

Specifically, Figures 1c, d, g, h and 2 c, d and 4 c, e were verified.

1)

As part of the model curation the ODEs were coded and verified with the authors. Specifically the initial conditions and the range of perturbations used to make figures 1c, g, were verified.

For simulation of the mutant in Fig. 1, the following ODEs were used:

$$\begin{aligned} z'[t] &== z[t] (y[t]/10 - 0.5) \\ y'[t] &== 0.25 (25 - (z[t] + zmut[t]) y[t]) \\ zmut'[t] &== zmut[t] (4 y[t]/10 - 0.5) \end{aligned}$$

with initial values:

$$\begin{aligned} z[0] &== 5 \\ y[0] &== 4 \\ zmut[0] &== 0, \text{ WhenEvent}[t == 10, zmut[t] -> 1] \end{aligned}$$

with the following queries, which were explained satisfactorily by the authors, and will be addressed in the final manuscript:

a) This results in a precise reproduction of Fig. 1d. It was for me not immediately clear how to incorporate the k-fold sensing mutant, described as: $zmut'[t] = zmut[t] * \lambda * (k * y)$ on line 507 of the manuscript. Both λ^+ and λ^- are dependent on y , as is stated on line 483 of the manuscript and it is not clear whether both λ^+ and λ^- should be multiplied by k . I tried both options and to reproduce the simulation result in the manuscript I should only multiply λ^+ with k (as shown in the equations above). I suggest to make this clear in the manuscript, specifically when one realizes that for the simulations in Fig. 1h both the λ^+ and the λ^- terms are affected by k (see below). It would appear that the y dependency of λ is changed by the mutation, i.e. $k * y$. The confusion stems from the statement that λ^- is y dependent on line 483, while this is not apparent in the equation on line 490.

b) It would be good to specify the initial value of the $zmut$ variable after the event of mutation. I used a value of 1, and this seems close to the value that you have chosen for the model simulations.

2)

For the biphasic circuit, I used the following equations:

$$\begin{aligned} z'[t] &== z[t] (4.8/(1 + (7/y[t])^5) - 6/(1 + (8/y[t])^5) - 0.1) \\ y'[t] &== 0.25 (25 - z[t] y[t]) \end{aligned}$$

with the same initial values and ranges as used for Fig. 1c, to reproduce Fig. 1g.

For the inclusion of the mutation event I used the following equations:

$$\begin{aligned} z'[t] &== z[t] (4.8/(1 + (7/y[t])^5) - 6/(1 + (8/y[t])^5) - 0.1) \\ y'[t] &== 0.25 (25 - (z[t] + zmut[t]) y[t]) \\ zmut'[t] &== zmut[t] (4.8/(1 + (7/(4*y[t]))^5) - 6/(1 + (8/(4*y[t]))^5) - 0.1) \end{aligned}$$

$$\begin{aligned} z[0] &== 6.1609 \\ zmut[0] &== 0 \\ y[0] &== 4.05785 \\ \text{WhenEvent}[t == 10, zmut[t] -> 1] \\ \text{WhenEvent}[t == 50, zmut[t] -> 1] \end{aligned}$$

Please confirm these equations, e.g. both $y[t]$ terms multiplied by $k=4$, and the steady state concentrations of $y[t]$ and $z[t]$ as initial conditions for the variables.

The authors confirmed the equations and soecified that the second event takes place at $t==47$, not at $t==50$.

3)

To simulate Fig. 4c and 4e, I used the following set of equations, which gave very comparable results to the results shown in the figures:

$$\begin{aligned} zs'[t] &== (2 pr - 1) lp zs[t] \\ zsm'[t] &== (2 prm - 1) lp zsm[t] \\ zd'[t] &== 2 (1 - pr) lp zs[t] + 2 (1 - prm) lp zsm[t] - lm zd[t] \end{aligned}$$

with initial conditions:

$$\begin{aligned} zs[0] &== 0.5 \\ zd[0] &== 1.0 \\ \text{and mutation event:} \\ zsm[0] &== 0.0, \text{WhenEvent}[t == 10, zsm[t] -> 0.01] \end{aligned}$$

and the following parameter values for the monophasic control:

$$\begin{aligned} lp &-> 1 \\ lm &-> 0.5 \\ pr &-> 1/(1 + \text{Sqrt}[k zd[t]]) /. k -> 1 \\ prm &-> 1/(1 + \text{Sqrt}[k zd[t]]) /. k -> 0.15 \end{aligned}$$

and for the biphasic control:

$$\begin{aligned} lp &-> 1 \\ lm &-> 0.5 \\ pr &-> (1/(1 + \text{Sqrt}[k zd[t]]) * 1/(1 + (1/(5 k zd[t])^4))) /. k -> 1 \\ prm &-> (1/(1 + \text{Sqrt}[k zd[t]]) * 1/(1 + (1/(5 k zd[t])^4))) /. k -> 0.15 \end{aligned}$$

with lp lambda+, and lm lambda-, prm the pr value for the mutant

Can you please confirm correctness of the equations, specifically $k=1$ for the wt and $k=0.15$ for the mutant and the initial conditions ($zs[0]$, $zd[0]$, and $zsm[0]$). I would recommend giving these values in the manuscript, for example in the supplementary material.

The authors confirmed the equations and indicated that k has a value of $1/6$ not 0.15 . They stated that this value and other parameter values will be added to the manuscript.

4)

I tried to code the model to simulate figures 2c and 2d, but could not find sufficient information in the manuscript to do so. Could you please send me a complete model description? I noticed references to a previous manuscript, but in that manuscript there was another reference for the model description. If you send me the full model description I can check the simulation results in the paper.

The authors submitted a complete model description, as given below, and stated that this will be added to the manuscript, including an error correction for lambda_minus.

Model description as submitted by authors:

```
dG/dt <- R0 - (EGO+SI*I)*G; // Glucose dynamics
dI/dt <- BETA*sigma*1/(1+(alpha/G)^1.7) + MBETA*sigma*1/(1+(alpha/(k*G))^1.7) - gamma*I;
// Insulin dynamics
```

```
dBETA/dt <- (1/(24*60))*BETA*(lambda_plus(G)-lambda_minus(G) - TAMOX) ;
dMBETA/dt <- (1/(24*60))*(MBETA*(lambda_plus(k*G)-lambda_minus(k*G)) +
BETA*TAMOX);
dTAMOX/dt <- (1/(24*60))*-1.5*log(2)*TAMOX
```

with:

```
lambda_plus(G) <- 0.1/(1+(8.4/G)^1.7)
lambda_minus(G) <- 0.2*(1/(1+(G/4)^8)+1/(1+(15/G)^6))
```

There was a mistake in the specification of lambda_minus in the supplementary information (will be fixed) - this is the lambda_minus we used for the simulations, which also corresponds to the set-point that we specified and to the function depicted in fig.2A.

The parameters that we used for the simulation:

```
alpha = 8.4;
sigma = 43.2 / (24*60);
gamma = 432 / (24*60);
R0 = 864 / (24*60) / (18);
EGO = 1.44 / (24*60);
SI = 0.72 / (24*60);
```

mutant scaling: k=6

initial values:

```
G[0] <- 4.966667;
I[0] <- 11.42;
BETA[0] <- 400;
MBETA[0] <- 0
TAMOX[0] <- 0.27
```

The model was simulated for t=40*24*60 minutes.

2nd Revision - authors' response

15 May 2017

Thank you very much for the positive consideration of our manuscript, for the reviewer comments and for the technical curation. We now added full specification of our models and simulations to the supplementary and methods sections so they can be readily reproduced, and addressed the issues pointed out by Prof. Snoep.

Technical curation for the mathematical models in MSB-17-7559R

The models described in the manuscript are clearly meant to be of a generic nature and not highly dependent on specific parameter values or initial conditions. However, to reproduce the figures in the manuscript it is necessary to have a full description of the model and it is MSB policy to have a full model description either in the manuscript or in supplementary material.

Below I give a summary of the e-mail communication with the authors, to clarify the model description in the manuscript.

With the additional information given by the authors, the model simulations given in the manuscript could be reproduced.

Specifically, Figures 1c, d, g, h and 2c, d and 4c, e were verified.

1)

As part of the model curation the ODEs were coded and verified with the authors. Specifically the initial conditions and the range of perturbations used to make figures 1c, g, were verified.

For simulation of the mutant in Fig. 1, the following ODEs were used:

$$\begin{aligned} z'[t] &== z[t] (y[t]/10 - 0.5) \\ y'[t] &== 0.25 (25 - (z[t] + zmut[t]) y[t]) \\ zmut'[t] &== zmut[t] (4 y[t]/10 - 0.5) \end{aligned}$$

with initial values:

$$\begin{aligned} z[0] &== 5 \\ y[0] &== 4 \\ zmut[0] &== 0, \text{WhenEvent}[t == 10, zmut[t] -> 1] \end{aligned}$$

with the following queries, which were explained satisfactorily by the authors, and will be addressed in the final manuscript:

*a) This results in a precise reproduction of Fig. 1d. It was for me not immediately clear how to incorporate the k-fold sensing mutant, described as: $zmut'[t] = zmut[t] * \lambda_{mut}(k*y)$ on line 507 of the manuscript. Both λ_{mut}^+ and λ_{mut}^- are dependent on y, as is stated on line 483 of the manuscript and it is not clear whether both λ_{mut}^+ and λ_{mut}^- should be multiplied by k. I tried both options and to reproduce the simulation result in the manuscript I should only multiply λ_{mut}^+ with k (as shown in the equations above). I suggest to make this clear in the manuscript, specifically when one realizes that for the simulations in Fig. 1h both the λ_{mut}^+ and the λ_{mut}^- terms are affected by k (see below). It would appear that the y dependency of λ_{mut} is changed by the mutation, i.e. $k*y$. The confusion stems from the statement that λ_{mut}^- is y dependent on line 483, while this is not apparent in the equation on line 490.*

Thank you for this correction, we added the following clarification in the relevant methods section (page 22 of the manuscript):

“Note that for the monophasic circuit simulated in Figure 1, the removal rate λ_- does not depend on y, and therefore it is not affected by the sensing mutation (only λ_+ is affected).”

b) It would be good to specify the initial value of the zmut variable after the event of mutation. I used a value of 1, and this seems close to the value that you have chosen for the model simulations.

We now added this to the relevant methods section (page 22):

“We simulated the invasion of a 4-fold sensing mutant in Figure 1DH by setting $Z_{mut} \leftarrow 1$ at specific time intervals in the simulation ($t=10$ for the monophasic circuit and $t=10, t=47$ for the biphasic circuit).”

We also added full specification of this model to the methods section, including initial values for the simulations (page 21-22):

“

Circuits with monophasic and biphasic control.

To simulate the circuits of Figure 1 in the main text, we used a circuit where a cell mass Z either increases the level of its input y (Fig. 1AE) or decreases the level of y (Fig. 1BF). The equation used for Z is:

$$\dot{Z} = Z \cdot (\lambda_+(y) - \lambda_-(y)) = Z \cdot \lambda(y) \quad [1]$$

Where λ_+ is the y -dependent proliferation rate of Z and λ_- is the y -dependent removal rate of Z .

In Fig. 1CDGH we simulated two cases – a monophasic circuit, where y increases the growth rate of Z , and a biphasic circuit, where y increases the growth rate of Z at low concentrations and decreases the growth rate of Z at high concentrations. The monophasic circuit was simulated using the growth rate equations:

$$\lambda_+(y) = \frac{y}{10} \quad [2]$$

$$\lambda_-(y) = 0.5 \quad [3]$$

and the biphasic circuit was simulated by using the growth rate equations:

$$\lambda_+(y) = \frac{4.8}{1 + \left(\frac{y}{5}\right)^5} \quad [4]$$

$$\lambda_-(y) = \frac{6}{1 + \left(\frac{y}{8}\right)^5} + 0.1 \quad [5]$$

These circuits were also used to simulate the phase plots in Figure 3AB. For Figure 3C we used the following circuit:

$$\lambda_+(y) = \frac{4.8}{1 + \left(\frac{y}{5.5}\right)^6} \quad [6]$$

$$\lambda_-(y) = \frac{6}{1 + \left(\frac{y}{6.3}\right)^6} + 0.3 \quad [7]$$

We used the following equation for the dependence of y on Z :

$$\dot{y} = \mu \cdot (M - Zy) \quad [8]$$

This equation means that Z increases the degradation rate of y , and at steady state we get $Z_{st}y_{st} = M$. We chose the parameters $M = 25, \mu = 0.25$.

Mutant invasion simulation.

We simulated the effect of a mutation by adding a term Z_{mut} such that:

$$\dot{y} = \mu \cdot (M - (Z + Z_{mut})y) \quad [1]$$

Z_{mut} represents the mass of cells with a (given) k -fold sensing mutation on y , so the growth rate of Z_{mut} is:

$$\dot{Z}_{mut} = Z_{mut} \lambda(ky) = Z_{mut} (\lambda_+(ky) - \lambda_-(ky)) \quad [2]$$

Note that for the monophasic circuit simulated in Figure 1, the removal rate λ_- does not depend on y , and therefore it is not affected by the sensing mutation (only λ_+ is affected). We simulated the invasion of a 4-fold sensing mutant in Figure 1DH by setting $Z_{mut} \leftarrow 1$ at specific time intervals in the simulation ($t=10$ for the monophasic circuit and $t=10, t=47$ for the biphasic circuit). The initial values for the simulations were $Z_{mut0} \leftarrow 0, Z_0 \leftarrow 5, y_0 \leftarrow 4$ for the monophasic circuit in Fig. 1D, and $Z_{mut0} \leftarrow 0, Z_0 \leftarrow 6.16, y_0 \leftarrow 4.06$ for the biphasic circuit in Fig. 1H.

2)

For the biphasic circuit, I used the following equations:

$$\begin{aligned} z'[t] &== z[t] (4.8/(1 + (7/y[t])^5) - 6/(1 + (8/y[t])^5) - 0.1) \\ y'[t] &== 0.25 (25 - z[t] y[t]) \end{aligned}$$

with the same initial values and ranges as used for Fig. 1c, to reproduce Fig. 1g.

For the inclusion of the mutation event I used the following equations:

$$\begin{aligned} z'[t] &== z[t] (4.8/(1 + (7/y[t])^5) - 6/(1 + (8/y[t])^5) - 0.1) \\ y'[t] &== 0.25 (25 - (z[t] + zmut[t]) y[t]) \\ zmut'[t] &== zmut[t] (4.8/(1 + (7/(4*y[t]))^5) - 6/(1 + (8/(4*y[t]))^5) - 0.1) \end{aligned}$$

$$\begin{aligned} z[0] &== 6.1609 \\ zmut[0] &== 0 \\ y[0] &== 4.05785 \\ \text{WhenEvent}[t == 10, zmut[t] -> 1] \end{aligned}$$

$WhenEvent[t == 50, zmut[t] \rightarrow 1]$

Please confirm these equations, e.g. both $y[t]$ terms multiplied by $k=4$, and the steady state concentrations of $y[t]$ and $z[t]$ as initial conditions for the variables.

The authors confirmed the equations and specified that the second event takes place at $t=47$, not at $t=50$.

We added full specification of this model to the methods section, including initial values for the simulations (see (1)).

3)

To simulate Fig. 4c and 4e, I used the following set of equations, which gave very comparable results to the results shown in the figures:

$$\begin{aligned} z_s'[t] &== (2 p_r - 1) l_p z_s[t] \\ z_{sm}'[t] &== (2 p_{rm} - 1) l_p z_{sm}[t] \\ z_d'[t] &== 2 (1 - p_r) l_p z_s[t] + 2 (1 - p_{rm}) l_p z_{sm}[t] - l_m z_d[t] \end{aligned}$$

with initial conditions:

$$z_s[0] == 0.5$$

$$z_d[0] == 1.0$$

and mutation event:

$$z_{sm}[0] == 0.0, WhenEvent[t == 10, z_{sm}[t] \rightarrow 0.01]$$

and the following parameter values for the monophasic control:

$$l_p \rightarrow 1$$

$$l_m \rightarrow 0.5$$

$$p_r \rightarrow 1/(1 + \text{Sqrt}[k z_d[t]]) /. k \rightarrow 1$$

$$p_{rm} \rightarrow 1/(1 + \text{Sqrt}[k z_d[t]]) /. k \rightarrow 0.15$$

and for the biphasic control:

$$l_p \rightarrow 1$$

$$l_m \rightarrow 0.5$$

$$p_r \rightarrow (1/(1 + \text{Sqrt}[k z_d[t]]) * 1/(1 + (1/(5 k z_d[t])^4))) /. k \rightarrow 1$$

$$p_{rm} \rightarrow (1/(1 + \text{Sqrt}[k z_d[t]]) * 1/(1 + (1/(5 k z_d[t])^4))) /. k \rightarrow 0.15$$

with l_p lambda+, and l_m lambda-, p_{rm} the p_r value for the mutant

Can you please confirm correctness of the equations, specifically $k=1$ for the wt and $k=0.15$ for the mutant and the initial conditions ($z_s[0]$, $z_d[0]$, and $z_{sm}[0]$). I would recommend giving these values in the manuscript, for example in the supplementary material.

The authors confirmed the equations and indicated that k has a value of $1/6$ not 0.15 . They stated that this value and other parameter values will be added to the manuscript.

We added full specification of this model to the methods section, including initial values for the simulations and parameter values (page 22-23):

“

Circuits of communicating stem cells.

In the study we presented two circuits that regulate the functional mass of differentiated cells, based on the model that is presented in Buzi et al. (Buzi *et al*, 2015). For the monophasic circuit, the equations are:

$$\dot{Z}_s = (2p_r(y) - 1)\lambda_+ Z_s \quad [1]$$

$$\dot{Z}_{smut} = (2p_r(ky) - 1)\lambda_+ Z_{smut} \quad [2]$$

$$\dot{Z}_d = 2(1 - p_r(y))\lambda_+ Z_s + 2(1 - p_r(ky))\lambda_+ Z_{smut} - \lambda_- Z_d \quad [3]$$

$$y \propto Z_d \quad [4]$$

where λ_+ is the stem cell division rate, λ_- is the differentiated cell removal rate, p_r is the probability that a stem cell that divided will not differentiate and $1 - p_r$ is the probability that it will differentiate. The population Z_{smut} is the population of stem cells with a k -fold sensing mutation. The monophasic replication rate $p_r(y)$, which is depicted in Figure 3B, was set as:

$$p_r(y) = \frac{1}{1 + \sqrt{y}} \quad [5]$$

The exact function used is not important, since as long as it is monotonically decreasing an invading mutant will take over. In the biphasic case, the replication rate used is:

$$p_r(y) = \frac{1}{1+\sqrt{y}} \cdot \frac{1}{1+\left(\frac{1}{5y}\right)^4} \quad [6]$$

The simulation of invading mutants is the same as for Figure 1 (which is explained in the mutant invasion simulation section). For the simulations we set $\lambda_+ \leftarrow 1$, $\lambda_- \leftarrow 0.5$, $k = \frac{1}{6}$ and with the initial conditions $Z_{s_0} \leftarrow 0.5$, $Z_{s_{mut_0}} \leftarrow 0$, $Z_{d_0} \leftarrow 1$. A mutation event was set such that $Z_{s_{mut}} \leftarrow 0.01$ at $t = 10$.

4)

I tried to code the model to simulate figures 2c and 2d, but could not find sufficient information in the manuscript to do so. Could you please send me a complete model description? I noticed references to a previous manuscript, but in that manuscript there was another reference for the model description. If you send me the full model description I can check the simulation results in the paper.

The authors submitted a complete model description, as given below, and stated that this will be added to the manuscript, including an error correction for lambda_minus.

Model description as submitted by authors:

```
dG/dt <- R0 - (EGO+SI*I)*G; // Glucose dynamics
dI/dt <- BETA*sigma*I/(1+(alpha/G)^1.7) + MBETA*sigma*I/(1+(alpha/(k*G))^1.7) - gamma*I;
// Insulin dynamics

dBETA/dt <- (1/(24*60))*BETA*(lambda_plus(G)-lambda_minus(G) - TAMOX) ;
dMBETA/dt <- (1/(24*60))*(MBETA*(lambda_plus(k*G)-lambda_minus(k*G)) +
BETA*TAMOX);
dTAMOX/dt <- (1/(24*60))*-1.5*log(2)*TAMOX
```

with:

```
lambda_plus(G) <- 0.1/(1+(8.4/G)^1.7)
lambda_minus(G) <- 0.2*(1/(1+(G/4)^8)+1/(1+(15/G)^6))
```

There was a mistake in the specification of lambda_minus in the supplementary information (will be fixed) - this is the lambda_minus we used for the simulations, which also corresponds to the set-point that we specified and to the function depicted in fig.2A.

The parameters that we used for the simulation:

```
alpha = 8.4;
sigma = 43.2/(24*60);
gamma = 432/(24*60);
R0 = 864/(24*60)/(18);
EGO = 1.44/(24*60);
SI = 0.72/(24*60);
```

mutant scaling: k=6

initial values:

```
G[0] <- 4.966667;
I[0] <- 11.42;
BETA[0] <- 400;
MBETA[0] <- 0
TAMOX[0] <- 0.27
```

The model was simulated for $t=40 \cdot 24 \cdot 60$ minutes.

We added full specification of this model to the appendix, including initial values for the simulations and parameter values (appendix section S3):

“

Blood glucose levels are regulated by the hormone insulin which secreted by pancreatic beta cells. The dynamics of glucose as a function of insulin can be described by the following minimal model (Bergman, 1989):

$$\dot{G} = u_0 + u(t) - (C + S_i I) \cdot G \quad [1]$$

where I is plasma insulin concentration, u_0 is endogenous production of glucose, $u(t)$ is meal intake, C is glucose removal rate at zero insulin and S_i is insulin sensitivity. Secretion of insulin is proportional to beta cell functional mass β and is modeled by the equation:

$$\dot{I} = p\beta \cdot \frac{G^{1.7}}{\alpha^{1.7} + G^{1.7}} - \gamma I \quad [2]$$

Where $\rho(G)$ is a monotonically increasing function of G , γ is the insulin removal rate and p is the insulin secretion per cell. Last, there is also a slow feedback where glucose controls the dynamics of beta cell proliferation and removal (Karin et al., 2016):

$$\dot{\beta} = \beta(\lambda_+(G) - \lambda_-(G)) = \beta \cdot \lambda(G) \quad [3]$$

The function $h(G)$ has a stable fixed point at $G = 5mM$. This slow feedback provides the system with robustness to variation in S_i, p since at steady state the dynamics of glucose to any input does not depend on these parameters (e.g. the system shows dynamical compensation (Karin et al., 2016)).

The function $h(G)$ also has an unstable fixed point at some $G \gg 5$, which results from glucose-dependant toxicity (glucotoxicity). This unstable fixed point can cause paradoxical beta cell death after an increase in glucose levels, which, in a self-reinforcing manner, further increases glucose levels. This process may underlie type 2 diabetes (De Gaetano et al., 2008; Ha et al., 2016; Karin et al., 2016; Topp et al., 2000). For our simulation, which is intended to represent young mice, we set this unstable fixed point to $G=13.5mM$ (Efanova et al., 1998; Maedler et al., 2006). The exact level of the unstable fixed point is not important for our conclusions, since a lower or higher unstable fixed point will work as well (as long as it is significantly smaller than $G=30mM$). We used the following function to model glucose dependent removal of beta cells:

$$\lambda_-(G) = \mu_- \cdot \left(\frac{1}{1 + \left(\frac{G}{4}\right)^8} + \frac{1}{1 + \left(\frac{15}{G}\right)^6} \right)$$

This death rate is similar to the glucose dependent death curve that is observed by Efanova et al (Efanova et al., 1998). Glucose dependent proliferation rate was modelled as in Karin et al (Karin et al., 2016):

$$\lambda_+(G) = \mu_+ \cdot \frac{1}{1 + \left(\frac{8.4}{G}\right)^{1.7}}$$

The values of μ_+, μ_- determine the turnover of beta cell functional mass and were set as:

$$\mu_+ = 0.1 \cdot \text{day}^{-1}$$

$$\mu_- = 0.2 \cdot \text{day}^{-1}$$

These values correspond to a ~3% turnover of beta cell functional mass per day. All other parameters of the βIG model were set as follows (Karin et al., 2016):

Parameter	Value	Units
u_0	$\frac{1}{30}$	mM min^{-1}
C	10^{-3}	min^{-1}
S_i	$5 \cdot 10^{-4}$	$\text{ml } \mu\text{U}^{-1} \text{min}^{-1}$
p	0.03	$\text{mg}^{-1} \mu\text{U ml}^{-1} \text{min}^{-1}$
α	8.4	mM
γ	0.3	min^{-1}

A beta-cell mutant with k -fold activation on the sensing of glucose has both a k -fold scaling of insulin secretion ($\rho(G) \rightarrow \rho(kG)$) and a k -fold scaling in its response in terms of growth rate ($\lambda(G) \rightarrow \lambda(kG)$). Therefore, to simulate the Y214C mutant (that has a 6-fold activation in glucose sensing) we simply replaced the secretion and growth functions accordingly, using $k = 6$. The combined equation for insulin secretion is the following:

$$\dot{I} = p\beta \cdot \frac{G^{1.7}}{\alpha^{1.7} + G^{1.7}} + p\beta_{mut} \cdot \frac{(kG)^{1.7}}{\alpha^{1.7} + (kG)^{1.7}} - \gamma I$$

Finally, in the experiment the Cre-mediated transgene was induced by tamoxifen. We simulated tamoxifen as converting normal beta cells to mutated beta cells:

$$\dot{\beta} = \beta(\lambda_+(G) - \lambda_-(G) - T)$$

$$\dot{\beta}_{mut} = \beta_{mut}(\lambda_+(kG) - \lambda_-(kG)) + \beta T$$

with T representing the concentration of tamoxifen in the blood. The dynamics of tamoxifen were simulated as exponential degradation with a half-life of 16 hours (Robinson et al., 1991) $\dot{T} = \frac{-\log(2)}{16 \cdot 60} T$.

The initial values used for the simulation:

Parameter	Value	Units
T	0.27	day ⁻¹
G	4.966667	mM
I	11.42	μU ml ⁻¹
β	400	mg
β_{mut}	0	mg

We simulated the dynamics of the system both by (i) assuming a quasi-steady-state for beta cell mass and solving equations [1],[2] to compute glucose levels, and (ii) explicitly modeling the dynamics of glucose and insulin using equations [1], [2], which adds a delay to the circuit. The model was simulated for $t = 40 \cdot 24 \cdot 60$ minutes. The results from (i) are provided in Fig. 1 in the main text and the results from (ii) are provided here as a supplementary figure (Appendix Figure S3). Because beta cell mass changes much slower than glucose, both methods yield highly similar results.

“

3rd Editorial Decision

22 May 2017

Thank you again for sending us your revised manuscript. We are now satisfied with the modifications made and I am pleased to inform you that your paper has been accepted for publication.

MODEL CURATION REPORT:

The authors have addressed all my queries with respect to model description adequately and have given complete model descriptions in the manuscript and supplementary material.

Corresponding Author Name: Prof. Uri Alon

Manuscript Number: MSB-17-7599